# Application of EU Geographical Indications for the Protection of Smoked Dry-Cured Ham—Impact on Quality Parameters

**DOI:** 10.3390/foods13244179

**Published:** 2024-12-23

**Authors:** Ana Gugić Ratković, Martina Turk, Helga Medić, Danijel Karolyi, Nives Marušić Radovčić

**Affiliations:** 1Faculty of Food Technology and Biotechnology, University of Zagreb, Pierottijeva 6, 10000 Zagreb, Croatia; ana.gugic.ratkovic@gmail.com (A.G.R.); martinaturk530@gmail.com (M.T.); hmedic@pbf.hr (H.M.); 2Faculty of Agriculture, University of Zagreb, Svetošimunska cesta 25, 10000 Zagreb, Croatia; dkarolyi@agr.hr

**Keywords:** dry-cured ham, smoking, EU geographical indications, PGI protection, quality, chemical and physical analysis, aroma, PAH compounds, sensory analysis, *Dalmatinski pršut*

## Abstract

This study compares *Dalmatinski pršut*, an EU Protected Geographical Indication (PGI)-labelled smoked dry-cured ham from the Dalmatia region in Croatia, with non-PGI hams from the same area, focusing on the impact of PGI certification on the product quality. The investigation is prompted by the increasing presence of dry-cured hams lacking the PGI label on the market, aspiring to compete with esteemed high-value PGI products. Samples of 28 smoked dry-cured hams (12 PGI and 16 non-PGI) were analysed for chemical properties, fatty acid composition, volatile compounds, PAH content and sensory characteristics. The results showed that PGI and non-PGI hams differ in their chemical and physical properties, with non-PGI hams having a lower water content and a higher salt content, which was also confirmed by the saltier taste in the sensory evaluation. PGI hams had a lower *b** colour value, and, while the differences in texture were minimal, non-PGI hams had slightly more monounsaturated fatty acids. The aroma analysis revealed that PGI hams contained more aldehydes and alcohols, while non-PGI hams had a higher content of phenolic compounds and aromatic hydrocarbons, probably due to differences in smoking practices. PAH levels, however, were within the EU limits, indicating controlled smoking of both type of hams. Overall, these results show that the quality of smoked dry-cured ham can be distinguished by the PGI designation. The study illustrates how the traditional production methods prescribed by the PGI production protocols shape the sensory and chemical profiles of *Dalmatinski pršut*, with the PGI certification playing a crucial role in maintaining product quality and certifying its typicality, to distinguish it from non-PGI products.

## 1. Introduction

Dry-cured ham, a globally renowned traditional meat product, stands out for its great palatability and distinctive organoleptic characteristics: a unique flavour profile and colour. However, the sensorial, physical–chemical, aromatic, morphological, and textural properties of dry-cured ham exhibit notable variations, due to differences in the technological processes of production [1] and various factors in pig production that can impact the quality of dry-cured ham [2]. Recognizing the need to preserve the authenticity and quality of such products, many distinguished dry-cured hams in the European Union (EU) have quality labels like Protected Designation of Origin (PDO) or Protected Geographical Indication (PGI). The authenticity guaranteed by these EU Geographical Indication (GI)-quality labels not only shields against misuse, imitation or unauthorized use, but also assures consumers of a genuine traditional product of recognized quality and local origin. These designations also ensure rigorous adherence to established quality procedures and controls. The EU GIs system, hence, serves as a guardian of the names associated with specific regions, safeguarding the unique qualities and reputations intrinsically tied to their geographical origins [3]. This protective framework is particularly vital in an era where consumers increasingly seek detailed information about product characteristics, place of origin, and production methods. GIs emerge as a response to this demand, acting as a shield that secures the connection between products and their geographical roots. Beyond meeting consumer expectations, GIs contribute significantly to territorial development by elevating the value of products and enhancing the competitiveness of local producers in domestic and international markets [4].

Croatia has a long tradition of producing local dry-cured hams, which is primarily associated with the Dalmatia region in the windy Adriatic Sea hinterland. Dalmatian dry-cured ham is traditionally prepared without the pelvic bone, and smoked, which gives it a unique flavour that is highly appreciated on the local market. Since 2016, Dalmatian dry-cured ham, labelled as *Dalmatinski pršut*, has PGI protection, so this product is often priced in a higher category, which adds considerable value to the local economy and plays a crucial role in increasing the producers’ income [5]. For instance, in 2022, the license to use the protected designation *Dalmatinski pršut* and the PGI label was utilized by eight medium-to-large-scale producers, who produced a total of 66,789 certified hams, which roughly corresponds to 15% of the annual production of all dry-cured hams in Croatia. In the case of *Dalmatinski pršut*, PGI protection ensures that the product is produced according to the traditional method, which includes meticulous steps such as ham trimming, dry salting, pressing, washing, smoking, air-drying and ripening in the specific geographical area. Compared to counterpart hams without a PGI, the traditional production process is based on a long smoking and drying process (up to 45 days), as provided for in the technical specification for a PGI product. In addition, the PGI product specification sets out several requirements for the quality of raw ham, relating to the minimum weight and freshness of the leg, the pH value of the meat and the coverage of fatty tissue, as well as the main physical–chemical parameters (e.g., maximal moisture and NaCl content, and water activity—aw) of the final product. However, the origin of raw hams is generally not specified, which can contribute to the variability of raw material and consequently affect the quality of the dry-cured product [2].

This study, therefore, aims to examine the quality attributes (chemical properties, fatty acid composition, volatile compounds, and sensory characteristics), as well as the safety (PAH content) of smoked dry-cured ham possessing the EU PGI label, specifically focusing on *Dalmatinski pršut*, in comparison to non-PGI-labelled smoked dry-cured hams available on the Croatian market. The investigation is prompted by the presence of dry-cured hams lacking the PGI label on the market, aspiring to compete with esteemed high-value products.

## 2. Materials and Methods

### 2.1. Samples

The research was carried out on 28 smoked dry-cured hams, collected from the National Fair of dry-cured hams in Sinj, Croatia. Samples were divided into two groups: PGI ham (protected geographic indication—PGI—smoked dry-cured ham *Dalmatinski pršut*) (N = 12) and non-PGI (smoked dry-cured ham without PGI protection) (N = 16). Heavy white pigs from a conventional production system were used for all ham processing. PGI hams were produced according to the PGI specification for *Dalmatinski pršut* [6]. In brief, raw back legs of the pig, weighing at least 11 kg and excluding the hooves but including the pelvic bones, skin, and subcutaneous adipose tissue, were used for salting. Salting using sea salt (Pag 91, Croatia) was conducted in cooling chambers at T 2–6 °C and RH of >80–90%, for a period of 14 to 20 days, depending on the weight of the raw ham. The salting phase was followed by a pressing phase, under the same conditions, for the period of 7–10 days. The hams were pressed into the typical shape, facilitating the penetration of salt. After pressing, the hams were washed with cold water and, after draining, hung in drying chambers with controlled microclimatic conditions (T 12–16 °C; RH reduced from 90 to 70%). During the drying phase (air flow rate 0.3 to 2 m/s), which lasted up to 45 days, the dry-cured hams were cold-smoked (T < 22 °C) by using cold smoke obtained by burning hardwood or the sawdust of beech (*Fagus* sp.), oak (*Quercus* sp.) or hornbeam (*Carpinus* sp.). The hams were then moved to a darkened cellar for ripening at stabile temperatures between 12 and 15 °C and RH between 65 and 75%. At the end of ripening (12 months) dry-cured hams were sampled for analysis. For the non-PGI samples, the production steps follow the same general process; however, these steps are not strictly controlled, which can lead to deviations from the technology specified for PGI hams. Samples of *biceps femoris* were coded, vacuum packed, frozen and kept at −18 °C until they were analysed. Sensory analysis was performed on dry-cured samples immediately after sampling.

### 2.2. Chemical and Physical Analysis

Moisture content and sodium chloride were measured using the official method [7], fat content was estimated following AOAC guidelines [8], and protein content was determined by the Kjeldahl method [9]. The non-protein nitrogen content was measured following the methodology outlined by Monin et al. [10]. The proteolysis index was determined as the ratio of non-protein nitrogen to total nitrogen, with the latter being analysed using the Kjeldahl method. Water activity was measured with a Testo 650 water activity meter (Testo Inc., New York, NY, USA). Colour measurements were performed using a Minolta CM-700d spectrophotometer (Minolta, Tokyo, Japan). Lightness (*L**), redness (*a**) and yellowness (*b**) were determined [11].

The extent of lipid oxidation was assessed by quantifying thiobarbituric acid reactive substances (TBARS) following the method described by Bruna et al. [12]. The results were reported as milligrams of malonaldehyde (MDA) per kilogram of sample.

Protein oxidation, as measured by the total carbonyl content, was evaluated by derivatization with 2,4-dinitrophenylhydrazine (DNPH), according to the method described by Armenteros et al. [13], and described in detail by Marušić Radovčić et al. [14].

Texture profile analysis (TPA) of the dry-cured ham samples was carried out in five replicates at room temperature, using a texture analyser (Ametek Lloyd Instruments Ltd., UK) with a 50 kg load cell and NexygenPlus 3.0 software. *Biceps femoris* muscle samples were cut into 10 × 10 × 10 mm pieces and conditioned at 20 °C for 2 h prior to testing. Each sample was compressed twice to 50% deformation, at a crosshead speed of 1 mm/s, with a 5 s resting time between cycles. The following parameters were derived from the force–distance curves: hardness (N), cohesiveness, gumminess (N), springiness (mm), chewiness (Nmm), and resilience.

### 2.3. Fatty Acid Composition

The fatty acid composition was analysed using gas chromatography. Fatty acid methyl esters (1 μL), prepared following the ISO method [15], were injected into an Agilent 6890N GC System (Santa Clara, CA, USA) equipped with a flame ionization detector (FID), as outlined in the ISO method [16]. The methyl esters were separated on a DB-23 column (60 m × 0.25 mm × 0.25 μm) (Agilent, Santa Clara, USA). Helium was used as the carrier gas at a flow rate of 1.5 mL/min. The injector temperature was set to 250 °C, and the detector temperature to 280 °C. The oven temperature was programmed to increase at a rate of 7 °C/min, starting from 60 °C and reaching 220 °C, where it was maintained for 17 min. The split ratio was set to 30:1. Fatty acid methyl ester (FAME) peaks were identified by comparing their retention times with those of FAME standards (C8–C22). The fatty acid composition results were expressed as a percentage of the total fatty acids in the dry-cured ham samples. Additionally, the percentages of total saturated (SFA), monounsaturated (MUFA), and polyunsaturated (PUFA) fatty acids were calculated.

### 2.4. Analysis of Volatile Compounds

An Agilent 6890N gas chromatograph coupled to a 5975i mass selective detector (Agilent Technologies, Santa Clara, CA, USA) was used in volatile compound analysis. Extraction of headspace volatile compounds was performed using solid-phase microextraction (SPME) with a 50/30 μm DVB/Carboxen/PDMS fibre for manual injection (Supelco, Bellefonte, PA, USA). Before the analysis, the fibre was preconditioned as indicated by the manufacturer (270 °C for 30 min). A total of 5 g of samples was homogenized with 25 mL distilled water saturated with NaCl in a commercial blender. Ten millilitres of this mixture was placed into 20 mL vials, and 100 μL of 4-methyl-2-pentanol (1.2 mg/kg) (internal standard) was added and tightly capped with a PTFE septum. Duplicate 20 mL vials were placed in a thermoblock at 40 °C. The SPME fiber was then exposed to the headspace for 180 min, while maintaining the sample at 40 °C. The analysis and calculation of retention indices (RIs) of volatile compounds by GC-MS was performed, following the method outlined by Marušić et al. [17].

### 2.5. Determination of PAH Compounds

The concentration of PAH compounds was determined by the method described by Bogdanović et al. [18]. A total of 1 g of homogenized *biceps femoris* muscle sample was saponified with ethanolic 2 N potassium hydroxide solution in a water bath (Seelbach, Germany) at 80 °C for 2 h. Liquid–liquid extraction with cyclohexane was performed to isolate PAHs containing an unsaponifiable fraction. Detailed sample preparation is described in the mentioned method [18]. The HPLC analysis of PAHs was conducted using an Ultra High Performance Liquid Chromatograph (Agilent 1290 Infinity UHPLC, Santa Clara, CA, USA) equipped with a binary gradient pump (G4220A) and an auto-sampler coupled with a fluorescence detector (Agilent 1260 Infinity Fluorescence Detector, Santa Clara, CA, USA). Compound separation was performed on a Hypersil Green PAH C18 analytical column (150 mm × 3 mm, particle size 3 μm; Hypersil™ Green PAH LC Column, Thermo Fisher Scientific, Waltham, MA, USA). Conditions of the analysis are explained in detail by Bogdanović et al. [18]. The PAH compounds were quantified by constructing external calibration curves for each individual PAH, which covered seven concentration levels ranging from 0.25 to 20 μg/kg.

### 2.6. Sensory Analysis

Sensory traits were evaluated by seven panel members, experienced in dry-cured ham analysis, using the quantitative descriptive analysis [19]. Evaluation of the 28 dry-cured ham samples was performed in 7 sessions (eight samples per session). Two replicates of each sample were analysed. All the samples were evaluated at 20–22 °C in sensory panel rooms. Samples from each manufacturer were labelled with random, three-digit codes, and presented on a plate at room temperature with water and bread without salt, to cleanse the palate between samples. The assessment of the whole slice of dry-cured ham (keeping 1 cm of subcutaneous fat) was made by evaluating sensory attributes. Sensory attributes were assessed on a 5-point line scale. Fat colour, red colour, marbling, odour intensity, flavour intensity, salty taste, sweet taste, smoky aroma and tenderness of dry-cured ham were evaluated using a 5-point line scale as described: fat colour (1 = yellow, 2 = light yellow, 3 = pale white, 4 = white, 5 = very white), red colour (1 = very light, 2 = light, 3 = moderate, 4 = deep, 5 = very deep red), marbling (1 = low presence of intramuscular fat, 2 = slightly low, 3 = moderate, 4 = high, 5 = very high presence of intramuscular fat), odour intensity (1 = low, 2 = slightly low, 3 = moderate, 4 = high, 5 = very high), flavour intensity (1 = low, 2 = slightly low, 3 = moderate, 4 = high, 5 = very high), salty taste (1 = not salty, 2 = slightly salty, 3 = moderately salty, 4 = very salty, 5 = excessively salty), sweet taste (1 = not sweet, 2 = slightly sweet, 3 = moderately sweet, 4 = very sweet, 5 = excessively sweet), smoky aroma (1 = low, 2 = bland, 3 = moderately bland, 4 = slightly intense, 5 = very intense) and tenderness (1 = very tough, 2 = moderately tough, 3 = slightly tough, 4 = just about right, 5 = very tender).

### 2.7. Statistical Analyses

A one-way ANOVA was performed on all data using SPSS Version 9.0 (SPSS, Chicago, IL, USA). Statistical significance was determined at *p* ≤ 0.05, with Tukey’s honest significant difference test applied for post hoc comparisons. A *p*-value of <0.05 was considered statistically significant.

## 3. Results and Discussion

### 3.1. Results of Chemical and Physical Analysis

Table 1 presents the results of physical and chemical parameters observed in the *biceps femoris* muscle of PGI and non-PGI smoked dry-cured hams. Analysis of these results reveals statistically significant differences (*p* < 0.05) in water and NaCl content, indicating that non-PGI samples displayed lower water and higher NaCl content, compared to PGI dry-cured hams. It is worth noting that the PGI specification for *Dalmatinski pršut* does stipulate certain chemical properties, such as a moisture content of between 40% and 55%, a water activity (a_w_) of less than 0.93 and a salt content (NaCl) of between 4.5% and 7.5%, and that all the PGI products analysed here were within these limits. Achieving the desired level of dehydration and water activity (aw) in the product is crucial for the shelf life and microbiological safety of dried meats [20]. In the present study, the water content exhibited a statistically significant difference (*p* < 0.05) between groups, with non-PGI hams (43.98%) displaying lower water content compared to PGI hams (48.82%), which indicates a higher degree of dehydration in non-PGI process. This could be due to the generally less controlled environmental conditions (e.g., higher T and lower RH) during the drying process of non-PGI hams, but also because of the use of leaner raw hams, as with PGI products the thickness of the subcutaneous fat with the skin on the outer part of freshly processed tight area, measured vertically below the head of the femur (*caput femoris*), should be at least 15 mm. The lack of a sufficient subcutaneous fat layer can increase the desiccation processes [21], and this was probably one of the contributing factors that led to higher water loss in hams without PGI, despite the likely shorter drying time. However, the water activity did not differ significantly between the processes, and averaged between 0.731 for hams with PGI and 0.763 for hams without PGI. Dry-cured hams contain a high proportion of high-biological-value proteins, because they contain essential amino acids in favourable ratios [22]. Since the body cannot synthesize these essential amino acids on its own, their intake through food is crucial [23]. The protein content in dry-cured ham is around 30 g/100 g, depending on the drying time and fat content in the ham [24]. Based on the results presented in Table 1, there was no statistically significant difference in the total protein content (*p* > 0.05) between PGI and non-PGI hams. The total protein content in the samples ranged from 32.74% to 34.57%, consistent with the findings of other authors in their study on smoked dry-cured ham (27.5–40.2%) [25,26]. The fat content ranged from 6.15% to 8.79% and did not exhibit a statistically significant difference (*p* > 0.05) between the two groups. Similarly, the TBAR value ranged between 0.45 and 0.53 mg MDA/kg per sample. These findings are consistent with TBAR results reported for other varieties of dry-cured ham produced within the same time frame, indicating comparability across different types of dry-cured ham [20,27].

Salt confers numerous benefits to dry-cured ham, including enhancing microbiological stability by lowering water activity, imparting a delightful savoury flavour, and facilitating the partial solubility and cohesion of myofibrillar proteins. However, excessive salt content acts as a notable inhibitor of most muscle proteases [28]. In this study, the salt content in samples of dry-cured ham exhibited a statistically significant difference (*p* < 0.05) concerning PGI protection: non-PGI hams displayed a higher NaCl content (8.31%), surpassing permissible limits for a PGI product. This emphasizes the crucial role of PGI protection and product conformity control in the effective regulation of NaCl content and other physicochemical properties of dry-cured ham. Furthermore, the elevated NaCl content adversely affected the sensory parameters of dry-cured ham, as well as the volatile aroma compounds, as discussed subsequently.

Protein carbonylation represents a significant chemical alteration during protein oxidation [28]. The assessment of carbonyl levels in samples using the dinitrophenylhydrazine (DNPH) method remains the predominant approach for evaluating protein oxidation in meat [29]. The results revealed no statistically significant difference in carbonyl concentrations among dry-cured hams (*p* > 0.05), with values of 20.87 PGI and 19.07 for non-PGI samples. Variances in carbonyl content observed by other research likely stem from differences in raw materials or nuances of the manufacturing processes, such as the inclusion of nitrites and ascorbic acid, known for their antioxidant properties [30].

No statistically significant difference in proteolysis index (*p* > 0.05) was found based on PGI protection (24.47% PGI vs. 30.01% non-PGI). Schivazappa et al. [31] reported a 29.4% proteolysis index in Italian Parma ham, aligning with our findings. In Spanish hams, Perez Santaescolástica et al. [32] noted a proteolysis index ranging from 31.10% to 38.59%, and Pugliese et al. [33] found a 25.8% proteolysis index in unsmoked dry-cured ham, *Kraški pršut*, consistent with our results. Variations in the proteolysis index across ham types may stem from differences in raw materials, salting process, ripening time, humidity, and temperature during production. Older hams, with lower salt content, showed a higher proteolysis index, as confirmed by other authors [33].

The colour is extremely important for the quality of dry-cured ham, because it not only plays an important role for the consumer, but can also be an important indicator of numerous changes that take place during the technological production process. The colour of the dry-cured ham depends on the water and fat content and the myoglobin concentration, but also on various external factors [23]. The results of the colour of smoked dry-cured ham are shown in Table 1. A statistically significant difference exists only for the *b** parameter between unprotected and PGI samples (*p* < 0.05). *L** values in the samples ranged from 48.11 to 54.79, and were significantly higher than the *L** values in Italian Parma ham (34.25) [34], Spanish Iberian ham (38.78) [35] and other Spanish types of dry-cured ham (34.80) [36]. The values for the *a** parameter were lower (3.48–6.70) than for Iberian ham (18.92) and other Spanish hams (15.55), and Italian Parma ham (12.08) [34]. For the *b** parameter, values ranged from 5.49 to 8.58, consistent with Iberian ham (7.59) but lower than Spanish hams (10.50) [36] and Italian Parma (2.38) [34]. The reason for the lower *a** parameter in *Dalmatinski pršut* is the use of only sea salt, without nitrites and nitrates that affect the cured-meat red-colour formation. However, the genotype and age of the animals may also contribute to the observed colour results, as Karolyi et al. [37] reported much lower *L** and higher *a** values than in the present study when using curing salt with NaNO_2_ and a standard smoking process in hams from a local, slow-growing pig breed.

The sensory attributes of dry-cured ham significantly influence consumers’ perceptions of product quality, and are primarily influenced by proteolytic and lipolytic breakdown processes occurring during dry-curing [38]. The correlation between proteolysis and texture evolution throughout the dry-curing process has been extensively documented, and the ripening phase has been identified as important, with the most pronounced impact on proteolysis, thus exerting considerable influence on the resultant meat texture. The Texture Profile Analysis (TPA) parameters of muscle *biceps femoris* of dry-cured hams are shown in Table 2. Statistical analysis revealed that PGI protection did not yield significant differences (*p* > 0.05) across all examined parameters, including hardness, cohesiveness, gumminess, springiness, chewiness and resilience. This lack of significant distinction in the textural attributes is unsurprising, given that the hams had more or less the same duration of ripening period.

### 3.2. Fatty Acid Composition

In the intramuscular fat (*biceps femoris*) of the analysed PGI hams (Table 3), saturated fatty acids (SFAs) accounted for 39.52%, monounsaturated fatty acids (MUFAs) for 53.67% and polyunsaturated fatty acids (PUFAs) for 6.81% of the total identified fatty acid esters. The hams without PGI had a similar PUFA content (5.98%) and a higher (*p* < 0.05) MUFA content (56.77%), which was mainly due to the higher proportion of C18:1c (52.34 vs. 48.88%, *p* < 0.01), while the SFA content tended to be lower (*p* = 0.08) than in the hams with PGI (37.25 vs. 39.52%).

A higher proportion of SFA, which is associated with higher adiposity in pigs [38], is favourable for the processing of dry-cured ham, due to the lower incidence of rancidity [20]. On the other hand, the higher degree of fat unsaturation, especially the PUFA content, which is inversely associated with the intramuscular fat content is unfavourable from a processing point of view, as the oxidative stability and sensory properties of meat products may be impaired [39]. Thus, the FA results (together with the TBARS results) suggest that the fat technological quality of PGI and non-PGI hams is generally similar, with the PGI hams possibly having a higher SFA content, which could be an advantage due to the greater stability of saturated fats, reducing the likelihood of oxidation and preserving the ham’s quality and texture. From a nutritional point of view, the FA profile of the intramuscular fat of the all dry-cured hams analysed corresponds broadly to the characteristic values for pork, whereby the polyunsaturated/saturated fatty acid ratio (PUFA/SFA) found (0.16 in PGI and 0.17 in non-PGI dry-cured hams) is below the recommended values (>0.4) [40,41] and the ratio of omega-6 to omega-3 of around 27 is typically too high (nutritionally desirable values are ≤4) [41].

Compared to other dry-cured hams from white pigs fed commercial feed, such as Spanish Serrano ham, which reported averages of 35–40% for SFA, 45–50% for MUFA, and 10–15% for PUFA, with omega-6/omega-3 ratios ranging from 14 to 16 [22], the present results show similar SFA and MUFA content, but differ significantly in PUFA levels and omega-6/omega-3 ratios. The FA composition of pork and dry-cured pork products is due to many factors, such as genetic traits, gender, meat and fat content and differences in rearing and feeding systems [42,43]. Therefore, many possible factors can be responsible for the observed differences in the FA profile.

### 3.3. Aroma Compound Analyses

Results of volatile aroma compounds in PGI and non-PGI smoked dry-cured hams are shown in Table 4. A total of 94 volatile compounds were identified in all samples, which were categorized into the following chemical groups: 23 aldehydes, 20 ketones, 18 phenolic compounds, 12 alcohols, 6 aromatic hydrocarbons, 4 aliphatic hydrocarbons, 3 esters, 3 nitrogen-containing compounds, 2 acids, 2 terpenes and 1 sulphur compound. PGI ham samples were characterized by a significantly higher proportion of aldehydes and alcohols, while non-PGI samples contained a higher proportion of aromatic hydrocarbons and phenols, which is probably due to the more intensive smoking process. The increased presence of aldehydes and alcohols in the PGI samples correlates with the increased levels of volatile aroma compounds associated with these groups, while the non-PGI samples had higher concentrations of volatile phenols and aromatic hydrocarbons. Statistical analysis confirmed significant differences (*p* < 0.05) between PGI and non-PGI samples within the groups of aldehydes, phenols, alcohols, aromatic hydrocarbons and terpenes.

Aldehydes, the main secondary products of lipid oxidation, significantly impact the aroma of dry-cured ham, with concentrations varying by production process and PGI status. Aldehydes have a low threshold, making them significantly influential in the flavour of dry-cured hams [44]. The study shows a significant difference in aldehyde content between PGI and non-PGI samples (*p* < 0.05), with PGI samples having a higher proportion (47.63%) compared to non-PGI (31.97%), likely due to more intense smoking in non-PGI samples, which increases the phenolic compounds that mask aldehydes. Linear aldehydes like hexanal, nonanal, and octanal dominate in both types, with hexanal being most abundant in PGI samples. Aromatic aldehydes, including benzaldehyde, are slightly higher in non-PGI samples, contributing a bitter-almond note, consistent with findings in Iberian ham, due to the longer curing process [6,23].

The proportion of phenolic compounds was also higher in non-PGI hams (34.38%) than in the PGI samples (13.28%), which can be attributed to the more intensive smoking process. Methoxyphenols, which are crucial for the aroma of smoked products, were significantly present in non-PGI samples, with 2-methoxyphenol being the most important compound. Compared to softwood, hardwood traditionally used for smoking contains higher levels of 2,6-dimethoxyphenol, which explains the higher flavour component in non-PGI ham [45].

Significant differences were observed in the alcohol content between PGI and non-PGI hams (*p* < 0.05), with PGI samples exhibiting higher levels (21.24%) compared to non-PGI samples (13.59%). Alcohols, mainly produced through lipolysis, proteolysis, and microbial activity, contribute distinctively to the aroma profile of dry-cured meats, especially branched alcohols arising from microbial degradation of branched aldehydes [26]. Higher concentrations of 1-heptanol, 1-octen-3-ol, benzyl alcohol, and 2-nonen-1-ol in PGI hams suggest that these compounds may contribute to their unique aroma, with 1-octen-3-ol adding a mushroom-like flavour and 1-octanol imparting fatty notes [6].

Ketones also play a significant role in the volatile profile of *Dalmatinski pršut*, with slightly higher concentrations reported in this study (10.52% in non-PGI and 10.17% in PGI samples) compared to previous research [6,26]. Although no statistically significant difference was observed (*p* > 0.05), ketones such as 2,5-octadienone, 2,3-dimethyl-2-cyclopenten-1-one, and 3-methyl-2-cyclopenten-1-one were present in high quantities. Ketones are predominantly generated through lipid oxidation, amino acid decomposition, and the influence of wood smoke during the curing and smoking process. Lipid oxidation produces various volatile ketones from unsaturated fatty acids, while amino acid breakdown, particularly through the Maillard reaction, also contributes to ketone formation. Additionally, wood smoke introduces compounds like ketones, which interact with the meat’s surface and contribute to the intense aroma characteristics that are essential to the ham flavour [46].

Four aliphatic hydrocarbons were identified, with no statistically significant difference between the samples (*p* > 0.05). Their proportion averaged 1.6%, which is consistent with the results of previous studies [6,26]. Two acids were identified: hexanoic acid and decanoic acid. The proportion of these acids showed no statistically significant difference (*p* > 0.05) between PGI and non-PGI samples. Three esters were identified: ethyl octanoate, ethyl decanoate and isopropyl myristate. The ester content did not differ significantly between the samples analysed (*p* > 0.05), with an average ester content of about 0.9%. The low ester content can be attributed to the antimicrobial effect of NaCl during the prolonged ripening process [47].

Terpenes are primarily associated with the addition of spices [6], and since only salt is used in the production of Dalmatian dry-cured ham, the terpene content remains low, as observed in this study. Two terpenes were identified: limonene and linalool. A statistically significant difference in terpene content was found between non-PGI and PGI samples (*p* < 0.05). In particular, the non-PGI samples contained a lower proportion of terpenes (0.48%) compared to the PGI samples (1.31%), with a significantly higher proportion of limonene in the PGI samples.

Among other compounds, dimethyldisulphide, a sulphur-containing compound, was also identified in the samples. The proportion of sulphur compounds in the non-PGI and PGI samples showed no statistically significant difference (*p* > 0.05). Dimethyldisulfide has been identified in several studies of Iberian ham [48,49,50,51,52]. The dimethyldisulphide levels are particularly influenced by the aging time, as it was only detected on the 230th day of the production process [53]. This indicates that the development of sulphur compounds takes place at certain stages of maturation, which are decisive for the overall flavour profile of the ham.

In conclusion, the distinct aroma profiles of PGI and non-PGI dry-cured hams result from complex interactions of volatile compounds influenced by traditional smoking and curing methods. These differences highlight the role of PGI production in shaping the aromatic profile, and emphasize the importance of PGI status in preserving the unique sensory qualities of *Dalmatinski pršut*.

### 3.4. Results of PAH Compounds

The results of the PAH compound concentration (μg/kg) in PGI and non-PGI dry-cured hams are shown in Table 5. Of the PAHs analysed, naphthalene (Nap), fluorene (Flu), acenaphthene (Acp), phenanthrene (Phen), anthracene (Ant), fluoranthene (Flt) and pyrene (Py) with up to four benzene rings (so-called light PAHs), as well as benz[a]anthracene (BaA), chrysene (Chr), benzo[b]fluoranthene (BbFA), benzo[k]fluoranthene (BkFA), benzo[a]pyrene (BaP), dibenz[a,h]anthracene (DbahA), benzo[ghi]perylene (BghiP) and Indeno[1,2,3—cd]pyrene (IP) with more than five benzene rings (so-called heavy PAHs) were determined, with the limits of detection (LOD) ranging from 0.02 (BaA and Chr) to 0.91 μg/kg (Acp). In both the PGI and non-PGI group, light PAHs were generally more present than heavy PAHs. The highest mean concentrations (μg/kg) within the light PAHs in PGI and non-PGI dry-cured hams were found for Phen (2.98 and 6.90) and Flt (1.86 and 2.83), followed by Acp (0.73 and 1.52), Nap (0.98 and 1.04) and Py (0.34 and 1.07), all of which were determined at higher levels in the non-PGI group. However, the difference between the groups was only statistically significant for Phen and Ant (*p* < 0.05), while for Acp and Py there was a tendency towards a statistically significant difference (*p* < 0.1).

Heavy PAHs, such as BaP, are generally more stable and toxic than light PAHs and potentially carcinogenic to humans at high intake levels [54]. Therefore, the cumulative concentration of four heavy PAHs: BaP, Chr, BaA and BbF, referred to as PAH4, in addition to the individual concentration of BaP, has been used as a primary indicator of PAH contamination in food [55]. The maximum level for BaP and the sum of PAH4 for smoked meat and smoked-meat products set by EU legislation are 2.0 μg/kg and 12.0 μg/kg, respectively [56]. In the present study, the heavy PAHs with the highest concentration were Dba, Chr and IP, reaching the mean values of 0.79 μg/kg, 0.56 μg/kg and 0.49 μg/kg, respectively, in the group of dry-cured hams without PGI. The concentrations of all heavy PAHs, including BaP and the sum of PAH4, were generally lower in the PGI hams. However, the differences between the groups were not statistically significant (*p* > 0.05). The BaP and PAH4 levels found in hams with PGI (0.09 μg/kg and 0.73 μg/kg) and in hams without PGI (0.25 μg/kg and 1.40 μg/kg) were within the permitted limits, and can therefore be considered safe for consumption. The BaP and PAH4 levels found in the present study are well below the levels reported for traditionally home-smoked dry-cured hams [57,58]. This indicates that the smoking process for PGI-labelled *Dalmatinski pršut*, but also for its non-PGI-labelled counterparts, is well controlled in practice, despite some minor differences in PAH profiles observed.

### 3.5. Results of Sensory Analysis

The results of sensory analysis of PGI and non-PGI samples are shown in Figure 1. Seven (7) out of nine (9) investigated sensory parameters showed statistically significant difference (*p* < 0.05) when comparing PGI dry-cured hams with non-PGI. Non-PGI dry-cured hams had a more pronounced smoky aroma and salty taste, as well as greater red-colour intensity than PGI dry-cured hams. A higher score for salty taste is consistent with the NaCl content. Non-PGI hams had higher red-colour intensity, which was consistent with the *a** value results (non-PGI dry-cured hams had slightly higher *a** values than PGI dry-cured hams, which may be due to the use of nitrite salts not allowed in PGI processing), and the stronger smoky aroma may be due to more exposure to smoking treatment or a different (e.g., cheaper) kind of wood used in non-PGI dry-cured hams. The sweet taste and fat colour were more pronounced in PGI ham, which is one of the sensory characteristics required by the PGI specification. PGI hams also had higher odour and flavour intensity than non-PGI dry-cured hams (*p* < 0.05). No significant differences were observed for marbling or tenderness between non-PGI and PGI hams (*p* > 0.05). The studies comparing the sensory characteristics of meat products, depending on whether the product bears the GIs label, are scarce. For instance, when the PGI “Cecina de León”, a dry-cured beef from the Spanish province of León, and the non-PGI “Cecina” were analysed, only minor differences were found between the sensory characteristics of the two products [59]. This was explained by the fact that both products are produced from the same pieces of meat, the production area is very small, and sometimes the same producer manufactures Cecina with and without the PGI label. Although the similar circumstances apply to the present study, the observed differences in sensory profile between the PGI and non-PGI products analysed here were more pronounced for some key sensorial qualities, such as the less-salty taste of the PGI hams.

## 4. Conclusions

In this study, the quality and safety characteristics (i.e., aw, PAH content) of PGI *Dalmatinski pršut* and non-PGI dry-cured ham from the same production area and season were compared. Significant differences were found in the water and NaCl content, with the non-PGI hams having lower water and higher NaCl values. The PGI *Dalmatinski pršut* met the required specifications for moisture and water activity. No significant differences were found in protein, fat or carbonyl content, although the non-PGI hams had a higher salt content, which affected the sensory properties. Differences in colour were mainly found in the parameter *b**, with the PGI hams having slightly higher values. Both hams with PGI and hams without PGI contained higher levels of light PAHs, with hams without PGI showing slightly higher concentrations, although all PAH levels were within the EU limits. The texture was similar, while the fatty acid profile of non-PGI hams showed a slightly higher MUFA content. In the aroma analysis, 94 volatile compounds were identified. The PGI samples had more aldehydes and alcohols, while the non-PGI hams contained more phenolic compounds and aromatic hydrocarbons, probably due to more intensive smoking. The sensory analysis revealed that non-PGI hams had a stronger smoky aroma, a higher salty taste, and a red colour, while PGI hams had a more pronounced sweet taste, a fatty colour, and a higher odour and flavour intensity. These results illustrate how the PGI status affects the quality of *Dalmatinski pršut*, and underline the role of traditional production methods in defining its unique characteristics and typicality, which distinguishes it from non-PGI products. This study shows that the quality of smoked dry-cured ham can be differentiated, depending on the PGI designation.

## Figures and Tables

**Figure 1 foods-13-04179-f001:**
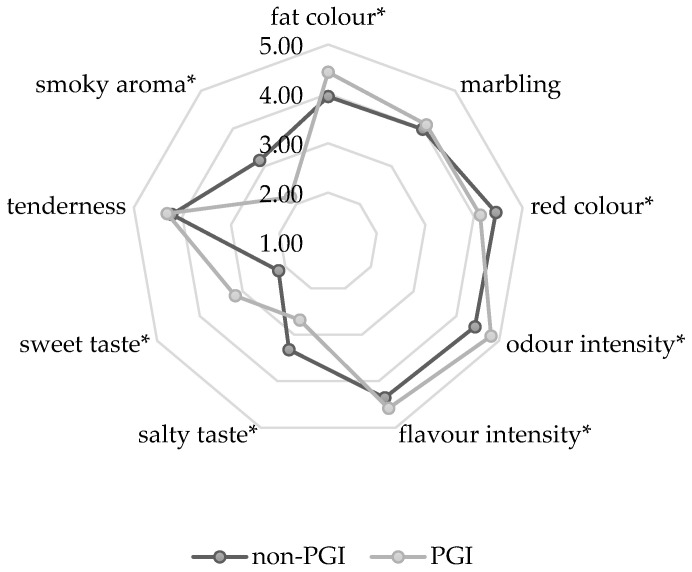
Sensory profile of PGI and non-PGI smoked dry-cured ham. The means marked with asterisk (*) are significantly different (Tukey’s test, *p* ≤ 0.05).

**Table 1 foods-13-04179-t001:** Physical and chemical parameters (mean ± standard error) in the *biceps femoris* muscle in PGI and non-PGI smoked dry-cured hams.

Parameter	PGI	non-PGI	*p*-Value
Water (g/100 g)	48.82 ± 0.96 ^b^	43.98 ± 1.18 ^a^	0.017
Fat (g/100 g)	6.15 ± 0.61	8.79 ± 0.98	0.104
Proteins (g/100 g)	32.74 ± 0.36	34.57 ± 0.80	0.095
NaCl (%)	7.33 ± 0.12 ^a^	8.31 ± 0.17 ^b^	0.001
a_w_	0.731 ± 0.02	0.763 ± 0.02	0.231
mg MDA/kg sample	0.53 ± 0.04	0.45 ± 0.02	0.102
nmol carbonyls/mg proteins	20.82 ± 1.22	19.07 ± 1.00	0.269
Proteolysis index	24.47 ± 1.61	30.01 ± 1.84	0.066
*L**	51.88 ± 0.43	49.98 ± 0.65	0.349
*a**	4.19 ± 0.39	5.10 ± 0.39	0.395
*b**	6.15 ± 0.29 ^b^	7.4 ± 0.32 ^a^	0.016

Different letters (a,b) within the same row indicate statistically significant difference (Tukey’s test, *p* ≤ 0.05).

**Table 2 foods-13-04179-t002:** Textural parameters in the *biceps femoris* muscle in PGI and non-PGI smoked dry-cured hams.

Texture Parameter	PGI	non-PGI	*p*-Value
Hardness (N)	53.98 ± 8.82	48.93 ± 4.33	0.386
Cohesiveness	0.52 ± 0.02	0.50 ± 0.03	0.746
Gumminess (N)	28.41 ± 5.48	24.26 ± 3.27	0.484
Springiness (mm)	−2.75 ± 0.33	−2.66 ± 0.22	0.955
Chewiness (Nmm)	67.95 ± 21.63	59.27 ± 10.62	0.631
Resilience	0.45 ± 0.04	0.40 ± 0.03	0.412

Results are expressed as mean ± standard error.

**Table 3 foods-13-04179-t003:** Fatty acid composition (% of total fat content) of the PGI and non-PGI smoked dry-cured hams.

Fatty Acid	PGI	non-PGI	*p*-Value
C10:0	0.14 ± 0.00 ^b^	0.10 ± 0.00 ^a^	0.000
C12:0	0.10 ± 0.00 ^b^	0.08 ± 0.00 ^a^	0.039
C14:0	1.49 ± 0.03 ^b^	1.29 ± 0.04 ^a^	0.003
C16:0	25.36 ± 0.45 ^b^	23.54 ± 0.51 ^a^	0.023
C16:1	3.56 ± 0.09 ^b^	2.98 ± 0.09 ^a^	0.001
C17:0	0.22 ± 0.01	0.21 ± 0.01	0.716
C17:1	0.24 ± 0.02	0.25 ± 0.01	0.970
C18:0	11.76 ± 0.21	11.57 ± 0.49	0.521
C18:1t	0.25 ± 0.01	0.30 ± 0.02	0.068
C18:1c	48.88 ± 0.35 ^b^	52.34 ± 0.85 ^a^	0.006
C18:2t	0.20 ± 0.00	0.22 ± 0.01	0.142
C18:2c	5.81 ± 0.68	5.03 ± 0.28	0.345
C18:3n3	0.27 ± 0.05	0.21 ± 0.02	0.234
C20:0	0.22 ± 0.02	0.23 ± 0.01	0.678
C20:1	0.74 ± 0.02 ^b^	0.90 ± 0.03 ^a^	0.001
C20:2	0.25 ± 0.04	0.28 ± 0.02	0.484
C20:4n6	0.28 ± 0.03	0.25 ± 0.01	0.439
C23:0	0.23 ± 0.05	0.21 ± 0.04	0.924
SFA	39.52 ± 0.57	37.25 ± 1.00	0.080
MUFA	53.67 ± 0.43 ^b^	56.77 ± 0.89 ^a^	0.015
PUFA	6.81 ± 0.78	5.98 ± 0.32	0.387
n6	6.09 ± 0.70	5.28 ± 0.29	0.344
n3	0.27 ± 0.05	0.21 ± 0.02	0.234
n6/n3	27.04 ± 3.17	27.60 ± 2.15	0.588
PUFA/SFA	0.16 ± 0.01	0.17 ± 0.0.1	0.620

Results are expressed as mean ± standard error. Different letters (a,b) within the same row indicate statistically significant difference (Tukey’s test, *p* ≤ 0.05).

**Table 4 foods-13-04179-t004:** Content of volatile compounds extracted in PGI and non-PGI smoked dry-cured hams (percentage of the total area).

Volatile Compound	RI	Kovats RI	PGI	non-PGI	*p*-Value
Aldehydes					
3-Methylbutanal	686	655 (642–666)	1.08 ± 0.63	0.56 ± 0.25	0.367
2-Methylbutanal	692	665 (640–670)	0.59 ± 0.17	0.55 ± 0.17	0.955
Pentanal	712	699	1.23 ± 0.33 ^b^	0.53 ± 0.18 ^a^	0.024
Hexanal	798	813 (796–812)	14.18 ± 1.62 ^b^	4.70 ± 2.19 ^a^	0.000
Heptanal	901	901 (899–907)	5.53 ± 1.97	2.48 ± 0.87	0.065
Benzaldehyde	966	963 (953–965)	2.78 ± 0.54	3.66 ± 1.73	0.802
Octanal	1004	1006	5.25 ± 2.19	3.67 ± 0.98	0.246
2,4-Heptadienal	1012	1000	0.13 ± 0.06	0.04 ± 0.06	0.190
Benzenacetaldehyde	1049	1044	3.32 ± 2.67	6.10 ± 4.90	0.425
2-Octenal	1062	1060	1.11 ± 0.26 ^b^	0.52 ± 0.43 ^a^	0.015
Nonanal	1105	1104	5.02 ± 2.45	4.18 ± 1.04	0.508
2-Methyl-2-heptenal	1114	1342	0.24 ± 0.15	0.43 ± 0.13	0.228
2-Nonenal	1162	1162	0.60 ± 0.20	0.36 ± 0.16	0.091
Decanal	1206	1209	0.38 ± 0.10	0.44 ± 0.16	0.793
2,4-Nonadienal	1214	1200	0.12 ± 0.05	0.07 ± 0.07	0.106
2-Decenal	1264	1250	0.75 ± 0.37	0.60 ± 0.24	0.392
Decadienal	1293	1297	0.17 ± 0.11	0.18 ± 0.16	0.958
2,4-Decadienal	1317	1325	0.47 ± 0.22 ^b^	0.12 ± 0.06 ^a^	0.044
Tetradecanal	1612	1618	0.11 ± 0.05	0.10 ± 0.07	0.634
Pentadecanal	1714	1711	1.19 ± 1.52	0.07 ± 0.07	0.325
Hexadekanal	1816	1819	2.87 ± 0.54	1.88 ± 0.79	0.090
9-Octadecanal	1995	1985	0.19 ± 0.06 ^b^	0.06 ± 0.03 ^a^	0.024
Octadecanal	2014	2021	0.07 ± 0.06	0.06 ± 0.07	0.780
		Total	47.63 ± 16.51 ^b^	31.97 ± 15.16 ^a^	0.024
Phenols					
Phenol	989	980	1.02 ± 0.90 ^b^	5.52 ± 1.97 ^a^	0.008
2-Methylphenol	1060	1072 (1042–1076)	0.78 ± 0.43 ^b^	2.48 ± 0.80 ^a^	0.018
4-Methylphenol	1081	1075 (1074–1093)	2.32 ± 1.53	4.20 ± 1.16	0.209
2-Methoxyphenol	1087	1088 (1087–1102)	4.25 ± 2.63 ^b^	11.83 ± 3.45 ^a^	0.005
2-Ethylphenol	1139	1146 (1138–1169)	0.08 ± 0.07	0.20 ± 0.15	0.169
2,4-Dimethylphenol	1150	1156 (1150–1169)	0.13 ± 0.12	0.28 ± 0.18	0.348
2,5-Dimethylphenol	1153	1153.5	0.09 ± 0.09	0.30 ± 0.19	0.255
3-Ethylphenol	1169	1169	0.10 ± 0.08	0.42 ± 0.20	0.060
3,5-Dimethylphenol	1172	1168 (1169–1172)	0.13 ± 0.09	0.34 ± 0.15	0.184
2-Methoxy-3-methylphenol	1175	1178	0.70 ± 0.73	0.36 ± 0.15	0.611
2,3-Dimethylphenol	1177	1185 (1181–1190)	0.01 ± 0.01	0.12 ± 0.09	0.093
2-Methoxy-5-methylphenol	1182	1191	1.67 ± 0.25	1.22 ± 0.25	0.113
2-Methoxy-4-methylphenol	1187	1181	0.81 ± 0.40 ^b^	3.72 ± 0.87 ^a^	0.000
3,4-Dimethylphenol	1192	1200 (1190–1200)	0.10 ± 0.05	0.30 ± 0.15	0.110
4-Ethyl-2-methoxyphenol	1273	1285	0.46 ± 0.23 ^b^	0.94 ± 0.19 ^a^	0.043
2-(1-Methylpropyl)-phenol	1313	1252	0.17 ± 0.07	0.10 ± 0.09	0.375
2,6-Dimethoxyphenol	1346	1348	0.47 ± 0.26 ^b^	1.85 ± 0.92 ^a^	0.007
		Total	13.28 ± 7.94 ^b^	34.38 ± 11.07 ^a^	0.001
Alcohols					
1-Penten-3-ol	705	686	0.33 ± 0.08	0.20 ± 0.12	0.403
3-Methyl-1-butanol	742	742	0.35 ± 0.18	0.14 ± 0.09	0.189
2.2-Dimethyl-3-heptanol	763	994	4.68 ± 1.41	5.74 ± 3.33	0.290
2-Furanmethanol	854	866	1.00 ± 0.59	1.55 ± 0.74	0.180
1-Hexanol	870	878 (858–888)	1.42 ± 0.95	0.51 ± 0.26	0.176
1-Buthoxy-2-propanol	950	948	0.24 ± 0.14	0.20 ± 0.15	0.715
1-Heptanol	980	968	1.63 ± 0.64 ^b^	0.59 ± 0.31 ^a^	0.046
1-Octen-3-ol	986	982	6.68 ± 2.37 ^b^	2.10 ± 1.22 ^a^	0.026
Benzyl alcohol	1039	1052 (1024–1051)	1.27 ± 1.36 ^b^	0.34 ± 0.16 ^a^	0.019
2-Nonen-1-ol	1073	1105	0.82 ± 0.23 ^b^	0.30 ± 0.15 ^a^	0.012
1-Octanol	1077	1083 (1064–1084)	2.47 ± 0.53 ^b^	1.42 ± 0.41 ^a^	0.012
Phenylethyl alcohol	1111	1118	0.35 ± 0.12	0.51 ± 0.26	0.378
		Total	21.24 ± 8.62 ^b^	13.59 ± 7.21 ^a^	0.027
Ketones					
2-Butanone	658	622	0.13 ± 0.02	0.12 ± 0.03	0.832
2-Heptanone	889	893(889–898)	0.78 ± 0.33	0.42 ± 0.14	0.141
3-Methyl-2-cyclopenten-1-one	969	973	0.43 ± 0.41 ^b^	1.56 ± 0.64 ^a^	0.010
3-Methyl-2(5H)-furanone	977	977	0.05 ± 0.05	0.13 ± 0.08	0.113
1-Octen-3-one	983	980	0.93 ± 0.63	0.33 ± 0.17	0.169
2,5-Octadienone	990	983	2.29 ± 1.12	1.43 ± 1.19	0.809
3,4-Dimethyl-2-cyclopenten-1-one	994	904	0.49 ± 0.35	0.83 ± 0.38	0.155
3,5-Dimethyl-2(5H) furanone	996	993	0.33 ± 0.21	0.41 ± 0.17	0.405
3-Methyl-1,2-cyclopentandione	1026	1043	0.20 ± 0.18	0.46 ± 0.22	0.250
2,3-Dimethyl-2-cyclopenten-1-one	1036	1035	1.30 ± 0.59 ^b^	2.24 ± 0.78 ^a^	0.039
5-Ethyldihydro-2(3H) furanone	1054	1056	0.13 ± 0.12	0.11 ± 0.08	0.820
Acetofenone	1066	1078 (1062–1068)	0.23 ± 0.11	0.37 ± 0.13	0.191
4,5-Dimethyl-4-hexen-3-one	1090	972	1.45 ± 1.36	0.58 ± 0.34	0.513
2-Nonanone	1093	1102 (1091–1104)	0.52 ± 0.38	0.16 ± 0.11	0.244
3,3-Dimethyl cyclohexanone	1134	1036	0.14 ± 0.15	0.08 ± 0.08	0.844
5-Buthyl dihydro-2(3H) furanone	1254	1255	0.12 ± 0.04	0.09 ± 0.04	0.342
2,3-Dihydro-1H-indenone	1276	1276	0.11 ± 0.07 ^b^	0.74 ± 0.25 ^a^	0.001
5-Pentyl-2(5H) furanone	1336	1337	0.16 ± 0.06 ^b^	0.05 ± 0.05 ^a^	0.033
Dihydro-5-pentyl-2(3H) furanone	1359	1363	0.11 ± 0.04	0.13 ± 0.05	1.000
6,10-Dimethyl-5,9-undecadien-2-one	1447	1448	0.08 ± 0.06	0.08 ± 0.06	0.689
		Total	10.17 ± 6.38	10.52 ± 5.10	0.370
Aromatic hydrocarbons					
Toluene	766	777 (758–780)	0.17 ± 0.08	0.30 ± 0.21	0.743
3-Methoxy pyridine	998	998	0.00 ± 0.00 ^b^	0.37 ± 0.21 ^a^	0.023
2,5-Dimethylfurane	1132	958	0.03 ± 0.03	0.08 ± 0.04	0.193
1,2-Dimethoxybenzene	1146	1147	0.32 ± 0.24 ^b^	0.99 ± 0.48 ^a^	0.012
3,4-Dimetoxytoluene	1237	1241	0.15 ± 0.10	0.45 ± 0.21	0.116
1,2,3-Trimethoxybenzene	1306	1315	0.05 ± 0.06 ^b^	0.77 ± 0.80 ^a^	0.017
		Total	0.72 ± 0.52 ^b^	2.96 ± 1.96 ^a^	0.001
Nitrogen compounds					
2,6-Dimethylpyrazine	914	914 (910–926)	0.25 ± 0.11 ^b^	0.77 ± 0.35 ^a^	0.026
2,3,5-Trimethylpyrazine	1000	1025 (1001–1014)	0.79 ± 0.31	0.77 ± 0.21	0.940
2-Methoxy-3-methylpyrazine	1126	972	0.28 ± 0.14 ^b^	0.53 ± 0.17 ^a^	0.035
		Total	1.31 ± 0.55	2.07 ± 0.73	0.059
Aliphatic hydrocarbons					
2,5-Dimethyl-1,4-hexadiene	908	1032	1.35 ± 0.82	1.45 ± 0.54	0.639
Dodecane	1200	1200	0.04 ± 0.04	0.08 ± 0.06	0.689
Tetradecane	1400	1400	0.05 ± 0.05	0.06 ± 0.04	0.964
Pentadecane	1500	1500	0.06 ± 0.06	0.06 ± 0.05	0.864
		Total	1.50 ± 0.96	1.75 ± 0.81	0.541
Acids					
Hexanoic acid	1007	1007	1.14 ± 1.21	0.59 ± 0.56	0.358
Decanoic acid	1371	1382 (1362–1387)	0.53 ± 0.25	0.62 ± 0.24	0.962
		Total	1.67 ± 1.46	1.21 ± 0.80	0.325
Esters					
Ethyl octanoate	1193	1193	0.16 ± 0.07	0.09 ± 0.07	0.168
Ethyl decanoate	1395	1395	0.08 ± 0.11	0.19 ± 0.13	0.503
Isopropyl myristate	1824	1824	0.72 ± 0.34	0.59 ± 0.64	0.858
		Total	0.96 ± 0.52	0.86 ± 0.84	0.893
Terpenes					
Limonene	1030	1033 (1027–1035)	1.14 ± 0.53 ^b^	0.38 ± 0.31 ^a^	0.035
Linalool	1100	1100	0.18 ± 0.09	0.10 ± 0.09	0.695
		Total	1.31 ± 0.62 ^b^	0.48 ± 0.40 ^a^	0.035
Sulphur compounds					
Dimethyldisulfide	747	747	0.20 ± 0.11	0.20 ± 0.07	0.766
		Total	0.20 ± 0.11	0.20 ± 0.07	0.766

Results are expressed as mean ± standard error. Different letters (a,b) within the same row indicate statistically significant difference (Tukey’s test, *p* ≤ 0.05). Method of identification: MS, RI.

**Table 5 foods-13-04179-t005:** Content (μg/kg) of PAH compounds (mean ± standard error) in PGI and non-PGI smoked dry-cured hams.

PAH Compound (μg/kg)	Abbrev	LOD ^1^	PGI	non-PGI	*p*-Value
Naphthalene	Nap	0.55	0.98 ± 0.09	1.04 ± 0.10	0.690
Fluorene	Flu	0.25	0.01 ± 0.01	0.00 ± 0.00	0.069
Acenaphthene	Acp	0.91	0.73 ± 0.18	1.52 ± 0.31	0.083
Phenanthrene	Phen	0.24	2.93 ± 0.43 ^a^	6.90 ± 1.32 ^b^	0.038
Anthracene	Ant	0.15	0.30 ± 0.13 ^a^	0.98 ± 0.22 ^b^	0.041
Fluoranthene	Flt	0.09	1.86 ± 0.32	2.83 ± 0.63	0.284
Pyrene	Py	0.26	0.34 ± 0.10	1.07 ± 0.30	0.089
Benz[a]anthracene	BaA	0.02	0.24 ± 0.11	0.32 ± 0.10	0.611
Chrysene	Chr	0.02	0.32 ± 0.13	0.56 ± 0.25	0.519
Benzo[b]fluoranthene	BbF	0.08	0.08 ± 0.03	0.19 ± 0.07	0.258
Benzo[k]fluoranthene	BkF	0.06	0.02 ± 0.01	0.03 ± 0.01	0.305
Benzo[a]pyrene	BaP	0.03	0.09 ± 0.05	0.25 ± 0.10	0.262
Dibenz[a,h]anthracene	Dba	0.23	0.12 ± 0.06	0.79 ± 0.32	0.132
Benzo[ghi]perylene	BghiPer	0.15	0.14 ± 0.05	0.19 ± 0.06	0.489
Indeno[1,2,3—cd]pyrene	IP	0.09	0.29 ± 0.10	0.49 ± 0.21	0.500
∑ BaA, Chr,BbF,BaP	PAH4 ^1^		0.73 ± 0.25	1.40 ± 0.58	0.414

Different letters (a,b) within the same row indicate statistically significant difference (Tukey’s test, *p* ≤ 0.05). LOD ^1^: limit of detection.

## Data Availability

The original contributions presented in this study are included in the article. Further inquiries can be directed to the corresponding author.

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
