# Peer review of "Application of EU Geographical Indications for the Protection of Smoked Dry-Cured Ham—Impact on Quality Parameters"

_foods, 2024, doi:10.3390/foods13244179_

Round 1
Reviewer 1 Report
Comments and Suggestions for Authors
In this study, the authors compared the quality of dry-cured ham with and without the EU Protected Geographical Indication (PGI) label. Overall, this study is interesting, the technique and methodology are appropriate. And the results can provide a theoretical basis and technical guidance for the industrial standardization and quality improvement of Dalmatinski pršut.
There are some specific comments or suggestions for the authors consideration, the following problems should be corrected:
Line 90: Hand legs?
Line 97: What are the parameters of the air circulation (e.g., flow rate) in the drying chambers?
Line 100: Regarding smoking woodchips, the composition of the woodchips has a significant impact on the physicochemical and sensory properties of the smoked meat product, so please provide a more specific and accurate composition of woodchips.
Line 118: the L*, a*, and b* should be italicized in the manuscript, as well as line 301-313.
Line 131: How many replicates of TPA for each group? Some texture parameters and units are not suitable for hams, such as adhesiveness, which was defined as the negative force area of the first bite, please refer to the Food Texture and Viscosity. Please check and revise this and ensure that other sections (R&D and Table 2) affected by this revision are also updated.
Line 159: Add the parameters (e.g., temperature and time) of precondition of fiber. And please choose one of the terms in whole paper: "fiber" in line 158, but "fibre" in line 159.
Line 160: How many volumes of solution were added to the sample homogenizes? Please add the detail.
Line 165-167: The statement is confusing, please revise.
Line 177: Italicize "biceps femoris".
Line 209: Regarding Sensory analysis, how many sensory sessions were held, "experienced in dry-cured ham analysis" is insufficient. The sensory attributes and descriptors are unclear, please provide specific scoring criteria for sensory attributes.
Additionally, an appropriate research ethics for sensory analysis should be mentioned with IRB numbers from the institute.
Line 211: ℃.
Line 222, 223, 229, and 237…: Please correct the form of p-value.
Line 335: How the authors explain why non-PGI hams have higher composition of C18:1c?
Line 358-360: The sentence should be rewritten, and "omega-3" in line 360.
Line 407: Ketone is also a predominant compound in smoked meat products, mainly generated from the lipid oxidization and amino acid decomposition, as well as wood smoke, please illustrate it.
Line 518-546: Conclusions: the statement is tedious. So, I would suggest condensing the sentences.
Reviewer 2 Report
Comments and Suggestions for Authors
Dear Autrhors,
Thank you for the submitted manuscript. The given issue and topic about hams, specially as traditional products such as Smoked -Dry cured Ham, are returning to our tables and are an integral part of the diet. I greatly appreciate the given article. Nevertheless, I have some minor issues about the given manuscript:
Abstract
The abstract describes the issue well. It is clear and introduces the reader to the issue. Nevertheless, I would add a little bit more information of the most significant results achieved in the present study.
Introduction
The Introduction is well written, I have no complaints about the section. I recommend checking the use of commas (,) in the text.
The end of the introduction describes the issue clearly and distinctly to the reader.
Material and methods
L 92 – add manufacturer of the sea salt
L96 – decribe pressing of the hams ? Not neceserally in the manuscript but I want to know how do you mean that sentene.
L106 – „..... frozen and stored at −18 °C until analysis.“ Try to change the words in the sentence
I dont have any other issues with decribed methods. Overal this part is well written and describes the study at an adequate level for manuscript.
Results and Discussion
L-237 – check the form of writting (p < 0.05) in the whole manuscript according to journal requierements
L287 – „No statistically significant difference in proteolysis.....“ - please add (p>0,05) to the sentence
L 350 – „... hams possibly having a higher SFA content, which could be an advantage.“ – why is this an andvatage ?
L378-390 – „Aldehydes, the main secondary products of lipid oxidation, have a significant impact 378 on the aroma of dry-cured ham, with concentrations varying depending on the produc- 379 tion process and PGI status. Present study indicates a statistically significant difference in 380 aldehyde content between PGI and non-PGI smoked ham samples (p < 0.05), with PGI 381 samples having a higher proportion (47.63 %) compared to non-PGI samples (31.97 %), 382 which is consistent with the findings of other authors [7,25]. This difference is probably 383 attributed to more intense smoking in non-PGI samples, which increases the phenolic 384 compounds that mask the aldehydes and thus decreases their proportion. Linear alde- 385 hydes such as hexanal, nonanal and octanal, which are formed by the oxidation of unsatu- 386 rated fatty acids such as oleic and linoleic acid, dominate in both sample types, with hex- 387 anal being the most abundant in PGI samples. Aromatic aldehydes, including benzalde- 388 hyde, are slightly higher in non-PGI samples, imparting bitter almond note, while they 389 are usually present in higher concentrations in Iberian ham due to the longer curing pro- 390 cess.“ – I would prefer this part a little bit shorter.
Conclusion
The conclusion is well written and explains the study's conclusions and recommendations for practical use.
Finally, I would like to thank you for the manuscript. I believe that the small changes I suggested will help improve your manuscript and it will be published.
Kind regards

Author Response
Reply to the Review Report (Reviewer 2)
Thank you very much for taking the time to review this manuscript. Please find the detailed responses below and the corresponding revisions marked in red in the re-submitted files.
Comments 1: Thank you for the submitted manuscript. The given issue and topic about hams, specially as traditional products such as Smoked -Dry cured Ham, are returning to our tables and are an integral part of the diet. I greatly appreciate the given article. Nevertheless, I have some minor issues about the given manuscript.
Response 1: Thank you for your kind words and for taking the time to review our manuscript. We are pleased to hear that you appreciate the topic of our study, particularly as it relates to traditional products like Smoked-Dry Cured Ham. We value your feedback and are happy to address the minor issues you’ve raised.
Comments 2: The abstract describes the issue well. It is clear and introduces the reader to the issue. Nevertheless, I would add a little bit more information of the most significant results achieved in the present study.
Response 2: Thank you for your comment. We have added additional information in the abstract about the water content and sensory differences between PGI and non-PGI ham found in the present study.
Comments 3: The Introduction is well written; I have no complaints about the section. I recommend checking the use of commas (,) in the text. The end of the introduction describes the issue clearly and distinctly to the reader.
Response 3: Thank you for your positive feedback on the Introduction section. We are glad to hear that it effectively presents the issue to the reader. We appreciate your suggestion regarding the use of commas, and we carefully revised the manuscript to ensure proper punctuation throughout (L18-19).
Comments 4: L 92 – add manufacturer of the sea salt
Response 4: Manufacturer of the sea salt was added. “(Pag 91, Croatia)” (L94)
Comments 5: L96 – decribe pressing of the hams? Not neceserally in the manuscript but I want to know how do you mean that sentene.
Response 5: The pressing stage in dry-cured ham production is a critical step aimed at improving the texture and shape of the ham while also aiding in the uniform distribution of salt. After the salting stage, the hams are pressed by stacking them in rows between plates and applying pressure (as you can see in the added picture). This process helps expel excess moisture from the ham, which is essential for proper curing and preservation. The pressing serves several purposes: it helps the ham retain its desired shape, contributes to the development of a firmer texture, and encourages the even penetration of salt.
Comments 6: L106 – „..... frozen and stored at −18 °C until analysis.“ Try to change the words in the sentence.
Response 6: Thank you for your comment. The sentence was changed to “Samples of biceps femoris were coded, vacuum packed, frozen and kept at −18 °C until they were analysed.”
Comments 7: L-237 – check the form of writting (p < 0.05) in the whole manuscript according to journal requierements
Response 7: Thank you for your comment. We have reviewed the manuscript and made the necessary changes to ensure that the formatting of statistical values, such as "p < 0.05," aligns with the journal’s requirements. All instances have been updated accordingly throughout the manuscript.
Comments 8: L287 – „No statistically significant difference in proteolysis.....“ - please add (p>0,05) to the sentence
Response 8: (p > 0,05) was added to the sentence. (L271)
Comments 9: L 350 – „... hams possibly having a higher SFA content, which could be an advantage.“ – why is this an andvatage ?
Response 9: The statement that hams possibly having a higher SFA content could be an advantage refers to the fact that saturated fats are more stable and less prone to oxidation compared to unsaturated fats. This stability can help maintain the quality and shelf-life of the ham, preventing the development of rancidity or off-flavors during storage. Additionally, higher SFA content may contribute to the desired texture and mouthfeel of the ham, as saturated fats tend to be solid at room temperature, providing a firmer, more cohesive structure. However, it is important to note that while SFA can offer these benefits, moderation and balance with other types of fats are recommended for overall health considerations.
So the sentence was changed to “Thus, the FA results (together with the TBARS results) suggest that the fat technological quality of PGI and non-PGI hams is generally similar, with the PGI hams possibly having a higher SFA content, which could be an advantage due to the greater stability of saturated fats, reducing the likelihood of oxidation and preserving the ham's quality and texture.“ (L332-336).
Comments 10: L378-390 – „Aldehydes, the main secondary products of lipid oxidation, have a significant impact 378 on the aroma of dry-cured ham, with concentrations varying depending on the produc- 379 tion process and PGI status. Present study indicates a statistically significant difference in 380 aldehyde content between PGI and non-PGI smoked ham samples (p < 0.05), with PGI 381 samples having a higher proportion (47.63 %) compared to non-PGI samples (31.97 %), 382 which is consistent with the findings of other authors [7,25]. This difference is probably 383 attributed to more intense smoking in non-PGI samples, which increases the phenolic 384 compounds that mask the aldehydes and thus decreases their proportion. Linear alde- 385 hydes such as hexanal, nonanal and octanal, which are formed by the oxidation of unsatu- 386 rated fatty acids such as oleic and linoleic acid, dominate in both sample types, with hex- 387 anal being the most abundant in PGI samples. Aromatic aldehydes, including benzalde- 388 hyde, are slightly higher in non-PGI samples, imparting bitter almond note, while they 389 are usually present in higher concentrations in Iberian ham due to the longer curing pro- 390 cess.“ – I would prefer this part a little bit shorter.
Response 10: This part is shortned as requested. Here you can find new version: “Aldehydes, the main secondary products of lipid oxidation, significantly impact the aroma of dry-cured ham, with concentrations varying by production process and PGI status. Aldehydes have a low threshold, making them significantly influential in the fla-vor of dry-cured hams [44]. The study shows a significant difference in aldehyde content between PGI and non-PGI samples (p < 0.05), with PGI samples having a higher propor-tion (47.63%) compared to non-PGI (31.97%), likely due to more intense smoking in non-PGI samples, which increases phenolic compounds that mask aldehydes. Linear al-dehydes like hexanal, nonanal, and octanal dominate in both types, with hexanal being most abundant in PGI samples. Aromatic aldehydes, including benzaldehyde, are slightly higher in non-PGI samples, contributing a bitter almond note, consistent with findings in Iberian ham due to the longer curing process [6,23].” (L364-374)
Comments 11: The conclusion is well written and explains the study's conclusions and recommendations for practical use.
Response 11: Thank you for your positive comment. However, another reviewer suggested that the conclusion be shortened, so we have made the necessary revisions to address this.
